# Germline Mutations in DNA Repair Genes in Patients with Pancreatic Neuroendocrine Neoplasms: Diagnostic and Therapeutic Implications

**DOI:** 10.3390/curroncol32110631

**Published:** 2025-11-10

**Authors:** Beata Jurecka-Lubieniecka, Małgorzata Ros-Mazurczyk, Aleksandra Sygula, Alexander J. Cortez, Marcela Krzempek, Anna B. Tuleja, Agnieszka Kotecka-Blicharz, Marta Cieslicka, Malgorzata Oczko-Wojciechowska, Daria Handkiewicz-Junak

**Affiliations:** 1Department of Nuclear Medicine and Endocrine Oncology, Maria Sklodowska-Curie National Research Institute of Oncology, Gliwice Branch, 44-102 Gliwice, Poland; 2Department of Clinical and Molecular Genetics, Maria Sklodowska-Curie National Research Institute of Oncology, Gliwice Branch, 44-102 Gliwice, Poland; 3Department of Biostatistics and Bioinformatics, Maria Sklodowska-Curie National Research Institute of Oncology, Gliwice Branch, 44-102 Gliwice, Poland; 4Digital Medicine Center, Maria Sklodowska-Curie National Research Institute of Oncology, Gliwice Branch, 44-102 Gliwice, Poland; 5Faculty of Medicine, Wroclaw Medical University, 44-367 Wroclaw, Poland

**Keywords:** pancreatic neuroendocrine neoplasms, pNEN, germline mutations, DNA repair genes, *BRCA2*, *CHEK2*, targeted sequencing, precision oncology, hereditary cancer, Polish cohort

## Abstract

**Simple Summary:**

Pancreatic neuroendocrine neoplasms (pNENs) are the second most common type of pancreatic cancer after ductal adenocarcinoma. While germline mutations in DNA repair genes are known to contribute to various hereditary and sporadic cancers, their role in pNENs remains unclear. This pilot study aimed to evaluate the frequency and clinical relevance of such mutations in Polish patients with pNENs, regardless of family cancer history. Germline DNA from 57 individuals was analyzed using targeted next-generation sequencing covering a panel of DNA repair genes. Mutations were found in 14 patients (24.6%). Pathogenic variants in *BRCA2* and *CHEK2* were identified in two cases, while seven patients carried variants of uncertain significance (VUS)**.** The detected alterations are associated with multiple malignancies, including breast, ovarian, prostate, gastric, colorectal, and pancreatic cancers. These findings suggest that germline DNA repair gene mutations may contribute to pNEN pathogenesis, even without familial predisposition. Broader germline testing and population-specific studies are warranted.

**Abstract:**

Pancreatic neuroendocrine neoplasms (pNENs) are the second most common type of pancreatic cancer after pancreatic ductal adenocarcinoma. Germline mutations in DNA repair genes drive several hereditary and sporadic cancers; however, their role in pNENs remains poorly defined. This pilot study aimed to assess the frequency and clinical relevance of germline DNA repair gene mutations in patients with pNENs, both with and without a family history of cancer. Germline DNA from 57 Polish patients with pNENs was analyzed using targeted next-generation sequencing to identify variants in a panel of DNA repair genes. Variant classification followed the American College of Medical Genetics and Genomics/Association for Molecular Pathology guidelines. Germline mutations were identified in 14 patients (24.6%), both with and without a family history of malignancy. Two patients carried pathogenic variants in *BRCA2* and *CHEK2*, while seven carried variants of uncertain significance (VUS). The identified variants have been implicated in various cancer types, including breast, ovarian, prostate, gastric, colorectal, and pancreatic cancers. These findings indicate that germline mutations in DNA repair genes may contribute to the pathogenesis of pNENs, even in patients without a family history. Broader germline testing and population-specific studies are needed to clarify the genetic landscape and clinical implications of these alterations.

## 1. Introduction

Pancreatic neuroendocrine neoplasms (pNENs) are the second most common type of pancreatic malignancy after pancreatic ductal adenocarcinoma (PDAC). Their incidence has increased nearly threefold over the past three decades, reaching 3–5 cases per 100,000 individuals annually. Although still classified as rare tumors, their prevalence is relatively high—comparable to colorectal cancer—owing to their typically indolent and prolonged clinical course [1,2]. This rising incidence largely reflects advances in imaging and molecular diagnostics, which have improved the detection of lesions with diverse biological characteristics [3,4]. pNENs arise from endodermal neuroendocrine cells of the pancreas, mainly within the islets of Langerhans, and exhibit pronounced molecular and clinical heterogeneity [5]. Functionally, some pNENs secrete bioactive hormones causing characteristic syndromes (e.g., insulinoma, gastrinoma, glucagonoma), whereas about 60% are non-functional and remain asymptomatic for years. Although most pNENs arise sporadically, a subset is associated with hereditary syndromes, including MEN1, VHL, NF1, and TSC [6]. Biologically, they span a spectrum from indolent, well-differentiated tumors to highly aggressive carcinomas. Consequently, the overall mortality rate remains around 50%, highlighting the need for improved diagnostic precision, molecular characterization, and stratified management of patients with pNENs [7].

Given this biological heterogeneity and the recognized role of germline alterations in cancer susceptibility, the exploration of DNA repair pathways represents a promising avenue for understanding pNEN pathogenesis. Defects in DNA repair mechanisms, which are crucial for maintaining genomic stability, are well-established contributors to both hereditary and sporadic cancers, including breast, ovarian, prostate, colorectal, and pancreatic adenocarcinomas [8,9,10]. Identifying germline mutations in key DNA repair genes has greatly advanced molecular diagnostics and enabled personalized therapeutic strategies [11,12].

According to the current National Comprehensive Cancer Network (NCCN) and European Society for Medical Oncology (ESMO) guidelines, germline testing is recommended for all patients with ovarian, breast, pancreatic, or prostate cancer—regardless of age or family history—to identify pathogenic or likely pathogenic variants in genes such as *BRCA1*, *BRCA2*, *PALB2*, *RAD51C*, and *RAD51D*. These recommendations highlight the importance of genetic counseling, cascade testing, and the therapeutic relevance of DNA repair defects, including the potential use of PARP inhibitors and other targeted agents in mutation carriers [13,14].

Unlike PDAC, where germline testing for DNA repair gene mutations has become standard practice, the germline genetic landscape of pNENs remains inadequately characterized [15,16,17,18]. This gap is partly attributable to the rarity and heterogeneity of pNENs [19,20]. However, recent progress in sequencing technologies has created new opportunities to explore their genetic basis [21,22]. A pivotal advance occurred in 2017, when the International Cancer Genome Consortium demonstrated, through whole-genome sequencing, the involvement of somatic mutations in DNA repair genes in the pathogenesis of pNENs and, for the first time, identified germline alterations [22]. Previously, such mutations were mainly linked to hereditary syndromes, including multiple endocrine neoplasia type 1, von Hippel–Lindau syndrome (VHL), neurofibromatosis type 1, and tuberous sclerosis complex, known to predispose individuals to pNENs [23,24]. Further evidence from U.S. cohorts supports the relevance of germline DNA repair mutations in pNENs, with frequencies of 18% in patients with a family history of cancer and 21% in those without [25]. A meta-analysis of 14 studies (221 patients) confirmed the consistent presence of germline mutations, particularly in familial cases, though the prevalence varied across cohorts [26]. Germline alterations have also been reported in young patients with neuroendocrine neoplasms and in poorly differentiated neuroendocrine carcinomas [27,28,29]. Moreover, the coexistence of pNENs with hereditary cancer syndromes involving DNA repair genes suggests a shared molecular basis [30,31,32,33]. Collectively, these findings highlight the need for broader genetic screening in patients with pNENs, irrespective of family history. They also raise several critical questions: Do germline mutations in DNA repair genes increase susceptibility to pNENs, as observed in other cancers? Can they serve as biomarkers to guide clinical decision-making and therapy? Should genetic counseling and surveillance be extended to family members?

Population-specific studies are also essential, as preliminary evidence suggests differences in mutational patterns between Caucasian and non-Caucasian cohorts, including Polish patients [Table A1]. This emphasizes the importance of regionally focused genetic research [23,34]. The frequency of the *CHEK2* del5395 variant in the Polish group (2%) is similar to values reported for the Finnish European population (2.576%), whereas it is far less common—below 0.1%—in African American and East Asian populations [https://gnomad.broadinstitute.org/, accessed on 18 June 2025]. Comparable population-specific patterns are observed for other tested genes, suggesting a unique genetic landscape in the Polish cohort and its relevance to cancer risk assessment.

This pilot study aimed to assess the frequency and clinical relevance of germline DNA repair gene mutations in a Polish cohort of patients with apparently sporadic pNENs, as well as in those with a family history of cancer. A better understanding of these alterations and their clinical implications could facilitate earlier detection, more tailored treatment strategies, and ultimately improved patient outcomes.

## 2. Materials and Methods

### 2.1. Patients

Patients were retrospectively identified through a review of medical records conducted between March and July 2024. Eligible individuals had been hospitalized between 2021 and 2024 at the Department of Nuclear Medicine and Endocrine Oncology, Maria Sklodowska-Curie National Research Institute of Oncology, Gliwice Branch. A total of 57 patients were included after application of strict inclusion and exclusion criteria.

### 2.2. Data Assessment and Patient Selection

Demographic, clinical, and pathological data were collected. Tumor staging followed the 8th edition of the American Joint Committee on Cancer (AJCC), and grading was based on the World Health Organization classification. Younger age at diagnosis, follow-up duration, and comorbidities were key evaluation parameters. A comprehensive family history was obtained, with particular attention to hereditary cancer syndromes and DNA repair gene mutations.

### 2.3. Informed Consent and Ethical Approval

All participants provided written informed consent after receiving a full explanation of the study aims and procedures. Ethical approval was obtained from the Bioethics Committee of the National Research Institute of Oncology, Gliwice Branch, in September 2021.

### 2.4. Inclusion Criteria

Histologically or cytologically confirmed diagnosis of pNEN between 2021 and 2024, verified at the reference center (National Research Institute of Oncology). Only well-differentiated tumors (G1–G2) were included.Signed informed consent form for participation.Availability of a detailed family cancer history.

### 2.5. Exclusion Criteria

Lack of histological confirmation of pNEN upon consultation.Medical conditions or abnormalities deemed by the study team to interfere with interpretation or compromise patient safety.Refusal to provide informed consent.

### 2.6. Molecular Analysis

The gene panel used in this study was designed based on previously published large-scale genomic analyses of pancreatic neuroendocrine neoplasms [22], allowing meaningful comparison across rare tumor cohorts. Supporting both molecular interpretation and potential therapeutic relevance. The selected 11 genes (*BRCA1*, *BRCA2*, *PALB2*, *CHEK2*, *MLH1*, *MSH2*, *MSH6*, *PMS2*, *EPCAM*, *APC*, *MUTYH*) were carefully chosen based on their well-documented role in DNA repair mechanisms, especially via homologous recombination and mismatch repair (MMR) pathways, which are crucial for maintaining genomic stability. This panel is routinely used in genetic diagnostics, which further supports its clinical relevance.

#### 2.6.1. DNA Isolation

Peripheral blood was collected in 5 mL BD Vacutainer^®^ tubes. Genomic DNA was isolated from leukocytes using the Maxwell^®^ RSC Blood DNA Kit on the Maxwell^®^ RSC Instrument (Promega GmbH, Madison, WI, USA). DNA concentration and quality were assessed using the Qubit^®^ 2.0 fluorometer (Thermo Fisher Scientific, Landsmeer, The Netherlands).

#### 2.6.2. Next-Generation Sequencing

Targeted next-generation sequencing (NGS) was performed on exons and flanking intronic sequences of the following genes: *BRCA1*, *BRCA2*, *PALB2*, *CHEK2*, *MLH1*, *MSH2*, *MSH6*, *PMS2*, *EPCAM*, *APC*, and *MUTYH*. Sequencing was performed using the Illumina MiniSeq platform (Illumina Inc., San Diego, CA, USA).

Library preparation, based on the Agilent SureSelect XT HS2 Custom Panel, included the following:Enzymatic DNA fragmentation (50–100 ng at 37 °C);End repair and A-tailing;Adapter ligation;Library amplification and purification;Target hybridization;Post-capture amplification.

Library quality was verified using the 4200 TapeStation system (Agilent Technologies, Santa Clara, CA, USA). Sequencing was performed using the paired-end method.

#### 2.6.3. Equipment Used

The following instruments and devices were used in sample preparation, library construction, and sequencing:NextSeq^®^ 550Dx sequencer (Illumina Inc.);Maxwell^®^ RSC Instrument (Promega GmbH);NanoDrop^®^ ND-1000 spectrophotometer (Thermo Fisher Scientific);Qubit^®^ 2.0 Fluorometer (Thermo Fisher Scientific);4200 TapeStation system (Agilent Technologies).

### 2.7. Bioinformatics Analysis

Sequence reads were aligned to the human reference genome using the Burrows–Wheeler Aligner (v 0.7.17) and Spliced Transcripts Alignment to a Reference aligner (v 2.7.11). Variant calling was performed using the Genome Analysis Toolkit (v 4.0.1.2) HaplotypeCaller and ExomeDepth. Variants were annotated and classified in accordance with American College of Medical Genetics and Genomics (ACMG) guidelines using Local Run Manager and Variant Interpreter (Illumina Inc., v 2.16). The threshold of coverage was set at 20× for small variants (SNPs, small indels) and 30× for CNVs. Quality Check was performed according to Illumina recommendation using Sequencing Analysis Viewer software (v 2.4.7).

### 2.8. Statistical Analysis

Categorical variables are summarized as frequencies and percentages. Group comparisons were performed using Fisher’s exact test or the Fisher–Freeman–Halton test. Continuous variables are reported as medians with interquartile ranges (IQR, 25–75%).

Normality of distributions was assessed using the Shapiro–Wilk test. Non-parametric comparisons between two groups were performed using the Wilcoxon rank-sum test. Genetic alterations and clinical features were visualized using oncoplots generated with the Complex Heatmap R package (v.2.10.0) [34]. All analyses were conducted using R version 4.4.2 (“Pile of Leaves,” released 31 October 2024; R Foundation for Statistical Computing, Vienna, Austria, http://www.r-project.org, accessed on 18 June 2025). A two-sided *p*-value < 0.05 was considered statistically significant.

## 3. Results

### 3.1. Study Group Characteristics

A total of 57 patients were included in the analysis: 14 (24.6%) carried germline mutations and 43 (75.4%) did not. The baseline clinical and demographic characteristics of the two groups are presented in Table 1.

A higher proportion of female patients was observed in the mutation group (79%) than in the non-mutation group (47%). Although this difference was not statistically significant (*p* = 0.062), a possible association between female sex and the presence of germline mutations was noted (odds ratio [OR] = 0.24; 95% confidence interval [CI]: 0.04–1.10).

The median age at the time of the study was similar between the two groups: 59 years (IQR, 52–65) in the mutation group and 61 years (IQR, 50–70) in the non-mutation group (*p* = 0.52). Likewise, the median age at diagnosis did not differ significantly (51 vs. 55 years; *p* = 0.48). The effect sizes (Cohen’s d) for both variables were small (*d* < 0.2), indicating negligible differences.

No significant differences in tumor grade distribution were observed between the groups. Similarly, no significant differences were observed in the tumor–node–metastasis classification. However, early-stage tumors (T1) were more common in the mutation group (50%) than in the non-mutation group (33%), while advanced stages (T2–T4) were less frequent.

Lymph node involvement (N1) and the presence of distant metastases (M1) were comparable between the groups (*p* = 0.76 and *p* = 0.75, respectively). Most tumors in both groups were hormonally non-functional, with no significant difference in endocrine activity (85.7% vs. 93%; *p* = 0.59).

Among comorbidities, cardiovascular disease was less frequent in the mutation group than in the non-mutation group (29% vs. 58%; *p* = 0.070), and this potential association warrants further investigation. The prevalence of diabetes did not differ significantly between the groups (*p* > 0.99).

No significant differences in the presence of other primary tumors or family history of malignancies were observed between the groups. Coexisting tumors (distinct from the primary diagnosis) were reported only in the non-mutation group (12%; *p* = 0.32). Family histories of breast, ovarian, prostate, or pancreatic cancer (50.0% vs. 42.0%; *p* > 0.99), as well as colorectal cancer and other malignancies, were comparable between the groups.

### 3.2. Genetic Findings

Targeted NGS identified two pathogenic variants and six variants of uncertain significance (VUS), according to ACMG criteria (Table 2).

Pathogenic variants:*CHEK2* c.(908+1_909-1)_(1095+1_1096-1)del (del5395)*BRCA2* c.8165C>G p.(Thr2722Arg)

VUS:*PMS2* c.2003T>C p.(Ile668Thr)*APC* c.3436C>T p.(Arg1146Cys)*MSH6* c.3242T>G p.(Leu1081Trp)*PALB2* c.3122A>C p.(Lys1041Thr)*MSH2* c.2178G>C p.(Met726Ile)*CHEK2* c.470T>C p.(Ile157Thr) (identified in four patients)

These variants affect pathways including mismatch repair (MMR) and homologous recombination repair of double-strand breaks (DSBs).

Table 2, Figure 1, and Table A2 present full variant details, patient characteristics, and an oncoplot visualization.

### 3.3. Interpretation of Genetic Variants

#### 3.3.1. Pathogenic Variants

*BRCA2* c.8165C>G p.(Thr2722Arg): This variant likely disrupts exonic splicing enhancers, leading to exon skipping and truncated protein function. Pathogenic *BRCA1/2* mutations are strongly linked to breast, ovarian, prostate, gastric, colorectal, and pancreatic cancers [35,36,37].*CHEK2* del5395: This deletion removes exons 9–10, impairing checkpoint kinase function in the DNA damage response. These mutations are associated with predisposition to breast, prostate, thyroid cancers, and thrombocythemia [38,39,40,41].

#### 3.3.2. Variants of Uncertain Significance (VUS)

*PALB2* c.3122A>C p.(Lys1041Thr): This variant has been previously observed in breast cancer and pediatric acute myeloid leukemia, though functional studies suggest limited impact [42].*CHEK2* c.470T>C p.(Ile157Thr): Classification varies from VUS to likely pathogenic. This variant may affect kinase signaling; however, population prevalence (up to 4.8%) complicates interpretation [43,44].*MSH2* c.2178G>C p.(Met726Ile): This conservative change has been reported in various cancers, but its clinical significance remains unclear.*MSH6* c.3242T>G p.(Leu1081Trp) and *PMS2* c.2003T>C p.(Ile668Thr): These are rare variants with limited evidence, and no functional data are available.*EPCAM* c.577A>G p.(Ile193Val): Although *EPCAM* inactivation can lead to *MSH2* silencing, this variant is not considered pathogenic based on current evidence.*APC* c.3436C>T p.(Arg1146Cys): This variant remains unclassified due to insufficient evidence.

#### 3.3.3. Benign Variants

Variants in *STK11*, *APC*, and *BRCA2* were classified as benign or likely benign based on their population frequency and lack of known functional impact.

### 3.4. Oncoplot: Genetic Alterations and Clinical Features Analysis

Analysis of the oncoplot revealed notable differences in both distribution and potential clinical relevance between patients harboring pathogenic variants and those carrying VUS. The most frequently mutated gene in the cohort was *CHEK2*, identified in 9% of patients, indicating its prominent role in hereditary cancer predisposition in this population. Recurrent alterations were also observed in *BRCA2* (5%), *PMS2* (4%), and *APC* (4%).

Less common alterations involved *MSH2*, *MSH6*, *PALB2*, *STK11*, and *EPCAM*, each detected in 2% of patients.

Genetic alterations occurred in both sexes; however, they were more frequent in female patients. The age at diagnosis was typically between 40 and 70 years, consistent with the expected age range for hereditary cancers. Among individuals with mutations in *CHEK2*, *BRCA2*, *PMS2*, and *APC*, a high proportion reported a family history of breast, ovarian, prostate, pancreatic, or colorectal cancer (8 out of 11 considered patients). Compared with the entire study cohort, these patients were more frequently diagnosed with tumors of higher histological grade G2 (n = 6, 55% vs. n = 16, 35%). In addition, distant metastases (M1) at the time of diagnosis were more common in this group (n = 6, 54% vs. n = 14, 32%). These findings support the notion that these mutations are associated with more aggressive tumor phenotypes and suggest a greater hereditary cancer burden among affected individuals. In contrast, patients carrying VUS, particularly in *CHEK2*, *BRCA2*, *PALB2*, and *EPCAM*, exhibited a more heterogeneous clinical profile. While some had a family history of cancer or coexisting risk factors, not all patients demonstrated aggressive tumor features. Furthermore, the disease course in this group appeared less uniform. This heterogeneity underscores the uncertain clinical impact of VUS and highlights the need for further functional and longitudinal studies to clarify their biological and prognostic significance.

## 4. Discussion

Mutations in DNA repair genes are increasingly recognized as important contributors to tumorigenic pathways in pNENs and may represent attractive therapeutic targets, particularly given the limited efficacy of currently available treatments [45,46]. Carcinogenesis in pNENs involves somatic mutations affecting critical cellular pathways, including PI3K/mTOR signaling, chromatin remodeling, telomere maintenance, and DNA damage repair [1,20,47]. Molecular studies have demonstrated both quantitative and qualitative diversity of somatic mutations within tumors, as well as distinct alterations in metastatic sites [34,48,49]. This heterogeneity remains a major obstacle in therapeutic decision-making [8,19,20,48].

Molecularly targeted therapies, including the mTOR inhibitor everolimus and the tyrosine kinase inhibitor sunitinib, are currently approved for patients with progressive or metastatic pNENs [50,51,52,53]. However, their efficacy varies considerably, largely due to the lack of validated biomarkers that could identify patients most likely to benefit [48]. This therapeutic uncertainty underscores the need for more personalized approaches grounded in a comprehensive understanding of both somatic and germline alterations.

DNA repair gene mutations represent one such avenue. However, their detection remains challenging because they may also occur in clinically silent pNENs. Recent large-scale genomic studies by Scarpa et al. and Ray et al. highlighted the significance of germline mutations in the tumorigenesis of pNENs [22,44]. Germline alterations, including pathogenic variants in *BRCA2*, *CHEK2*, and *MUTYH*, were detected in 10% of a cohort of 102 patients. Notably, such mutations were also found in patients with apparently sporadic pNENs and no family history suggestive of hereditary cancer syndromes [22]. Similarly, Mohindroo et al. identified pathogenic or likely pathogenic germline variants in 14% of patients, involving *CHEK2*, *ATM*, and *PALB2*, with a comparable prevalence among patients with and without a family history [25].

Germline mutations in DNA repair genes were identified in approximately 25% of our cohort of 57 patients with clinically sporadic pNENs. The prevalence of mutations did not differ significantly between patients with and without a family history of cancer. These findings have several important implications. First, they suggest that germline predisposition may play a more prominent role in the etiology of pNENs than previously recognized. Although historically considered sporadic, a significant subset of pNENs may arise from hereditary cancer syndromes. Because these genes often act recessively, cancer predisposition results from the inheritance of a single germline mutation, followed by loss of heterozygosity and biallelic inactivation [54]. Identification of carriers or diagnosis during the long asymptomatic phase preceding tumor progression is therefore crucial for early detection [14,55]. NCCN guidelines now recommend germline testing in patients with pNENs; for patients with confirmed familial mutations, testing should also be extended to relatives [14]. Surveillance of asymptomatic carriers facilitates early-stage detection, potentially enabling curative surgical intervention and reducing mortality in this group.

Second, pathogenic variants in *BRCA* and *CHEK2*, as well as VUS in MMR genes identified in this study, are associated with well-established cancer predisposition syndromes. Thus, these genetic alterations warrant careful clinical consideration. Germline *BRCA* mutations increase the risk of breast, ovarian, and prostate cancers, as well as PDAC. *BRCA1/2* are highly penetrant genes essential for homologous recombination repair, and their inactivation results in genomic instability and uncontrolled cell proliferation [56,57,58,59,60]. Of note, Scarpa et al. reported a germline *BRCA2* mutation in pNENs, some of which were identical to variants observed in PDAC [22]. In our series, a distinct pathogenic *BRCA* mutation was identified in a patient without a notable family history, whose son was recently diagnosed with PDAC, suggesting a broader shared molecular basis. Although several genes such as *BRCA1*, *XRCC4/6*, and *FANCM* are established biomarkers in pancreatic ductal adenocarcinoma (PDAC), they were not included in the present analysis, as the pathogenic *BRCA2* variant identified in our cohort represents the first and only molecular feature linking pNENs to PDAC biology. However, data remain limited, several studies have reported pathogenic *BRCA* mutations in isolated pNEN cases [61,62], underscoring the contribution of germline alterations even in apparently sporadic tumors. Germline *CHEK2* mutations have also been implicated in predisposition to breast and prostate cancer [63,64,65,66]. Scarpa et al. were among the first to report pathogenic *CHEK2* mutations in patients with apparently sporadic pNENs, predisposing them to prostate cancer [22]. In our cohort, a pathogenic *CHEK2* mutation was identified in one female patient. This variant is associated with an increased risk of familial breast and thyroid cancer; however, neither the patient nor her relatives had a history of these malignancies. Published reports have also documented cases of co-occurrence of breast cancer with *CHEK2* mutations and pNENs in two sisters [67]. Another *CHEK2* VUS was detected in several of our patients with family histories of breast, prostate, pancreatic, or lung cancer. The clinical interpretation of *CHEK2* VUS remains uncertain; such variants may complicate rather than facilitate genetic counseling, underscoring the need for further functional characterization to clarify their clinical significance [68,69].

Defective MMR is another relevant pathway. MMR deficiency leads to microsatellite instability and predisposes to several cancers, particularly colorectal and endometrial cancer [70]. Reduced expression of *MLH1* and *MSH2* proteins in sporadic pNENs has been reported [71,72], as well as germline mutations in *MLH1*, *MSH2*, and *MSH6* [73,74]. In our series, germline mutations in *MSH2*, *MSH6*, and *PMS2* were identified. In some cases, these mutations were associated with a family history of colorectal cancer, consistent with reports of pNENs in non-classical Lynch syndrome [75].

### 4.1. Clinical Translational Outlook

The identification of germline and somatic alterations in DNA repair genes among pNEN patients opens new avenues for translational research and future clinical application. Although current therapeutic strategies for pNENs primarily rely on somatostatin analogs, targeted agents (e.g., everolimus, sunitinib), and peptide receptor radionuclide therapy. Therapies such as platinum-based regimens, PARP inhibitors, and immune checkpoint blockade have been proposed for selected neuroendocrine tumors; these approaches remain investigational in pNENs and are currently under evaluation in early-phase trials only (ENETS 2023; ESMO 2024; NCCN 2025) [13,14]. In other solid tumors, *BRCA1/2*, *PALB2*, or *RAD51*-mutated cancers have demonstrated sensitivity to platinum-based chemotherapy and PARP inhibition, supporting the rationale for evaluating similar approaches in pNENs [76,77,78,79]. Furthermore, MMR-deficient and microsatellite instability–high (MSI-H) tumors—although rare in pancreatic neuroendocrine neoplasms—could benefit from immune checkpoint blockade with agents such as pembrolizumab or dostarlimab, as approved in tumor-agnostic indications [80]. Moreover, mechanism-based therapies are being introduced in genetically defined subgroups, as illustrated by the activity of the HIF-2α inhibitor belzutifan in VHL-associated neoplasms, including non-metastatic pNENs [81,82]. Future studies should aim to integrate molecular profiling into the clinical trial design for pNENs to identify patients most likely to benefit from DNA damage-targeted or immunotherapeutic strategies. Collaborative efforts across genomic and clinical consortia may help establish biomarker-driven treatment algorithms and clarify the prognostic and predictive value of specific germline variants. In summary, expanding molecular diagnostics to include DNA repair gene panels in pNENs could facilitate both earlier detection of hereditary syndromes and a more individualized therapeutic approach, bridging the gap between genetic insights and clinical management.

### 4.2. Clinical Significance

In our cohort, germline mutation carriers were more frequently female, more often diagnosed with T1 tumors, and less likely to have cardiovascular comorbidities. Some carriers of *CHEK2*, *BRCA2*, *PMS2*, and *APC* mutations exhibited features of aggressive disease, including higher grade, larger size, and metastases. Although these differences were not statistically significant, they align with previous reports suggesting that germline mutation carriers may have a distinct, and in some cases more aggressive, clinical course. Nonetheless, identification of such mutations provides an opportunity for early detection and preventive strategies. Larger studies are warranted to confirm these findings.

### 4.3. Limitations

This study has several limitations. First, the number of patients was relatively small, which limits statistical power and reduces the ability to draw definitive conclusions regarding the associations between specific germline mutations and clinical features. Second, the genetic analysis focused on selected DNA repair genes, which may not capture the full mutational landscape of pNENs. Given the rarity of pNENs, focusing on well-differentiated cases (G1–G3) allows meaningful aggregation and comparison of genomic findings across international datasets, including the ICGC reference cohort by Scarpa et al., thereby enhancing the interpretability and translational relevance of the results. Third, a proportion of the detected variants were classified as VUS, limiting their immediate applicability in clinical practice. Fourth, although the mean follow-up was 8.5 years, the observational design and limited sample size limit the ability to definitively assess the impact of germline mutations on disease progression, treatment response, or overall survival. Finally, the findings were not validated in an independent cohort and should be interpreted with caution until confirmed by larger, multicenter studies.

## 5. Conclusions

The germline landscape of pNENs includes clinically relevant mutations in DNA repair pathways, paralleling those observed in other solid tumors. Such mutations contribute to increased susceptibility to pNENs and may serve as biomarkers with therapeutic implications, supporting systematic germline testing for all patients. Pathogenic *BRCA* variants indicate potential sensitivity to platinum-based chemotherapy and PARP inhibitors, while MMR deficiency provides a rationale for immune checkpoint blockade. Detection of germline mutations also justifies genetic counseling and surveillance for family members, enabling early diagnosis and preventive strategies. In our cohort, germline mutation carriers were more often female, more frequently diagnosed at an early stage, and less likely to have cardiovascular comorbidities; however, these differences were not statistically significant. These findings highlight the importance of integrating germline testing and genetically informed therapies into the management of pNENs. Nevertheless, larger multicenter studies are required to validate these observations and refine precision medicine strategies.

## Figures and Tables

**Figure 1 curroncol-32-00631-f001:**
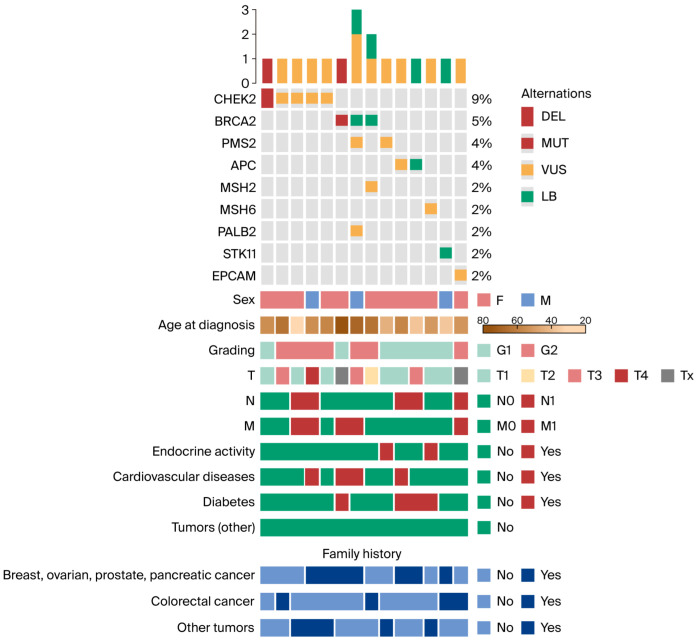
Oncoplot depicting genetic alterations and clinical features. Only patients harboring at least one mutation are shown. Abbreviations: DEL, deletion; MUT, mutation; VUS, variant of uncertain significance; LB, likely benign; T, tumor; N, node; M, metastasis; F, female; M, male.

**Table 1 curroncol-32-00631-t001:** Baseline clinical and demographic characteristics of patients with and without germline mutations.

Variable	Group	No Mutation (n = 43)	Mutation (n = 14)	*p*	Effect Size (95% CI)
Sex, n (%)	Female	20 (47%)	11 (79%)	0.062 ^F^	0.24 (0.04–1.1) ^OR^
	Male	23 (53%)	3 (21%)		
Age, median (Q1–Q3) (y)		61 (IQR 50–70)	59 (IQR 52–65)	0.52 ^W^	0.16 (−0.46–0.77) ^d^
Age at diagnosis, median (Q1–Q3) (y)		55 (45–63)	51 (45–60)	0.48 ^W^	0.20 (−0.42–0.82) ^d^
Grading, n (%)	G1	24 (56%)	7 (50%)	0.53 ^F^	1.59 (0.39–6.54) ^OR^
	G2	15 (35%)	7(50%)		
	Gx	4 (9%)	0 (0%)	–	–
T, n (%)	T1	14 (33%)	7 (50%)	0.42 ^FFH^	0.23 (0.11–0.52) ^φc^
	T2	11 (26%)	1 (7%)		
	T3	12 (28%)	3 (21%)		
	T4	3 (7%)	1 (7%)		
	Tx	3 (7%)	2 (14%)	–	–
N, n (%)	N0	22 (51%)	9 (64%)	0.76 ^F^	0.77 (0.17–3.16) ^OR^
	N1	16 (37%)	5 (36%)		
	Nx	5 (12%)	0 (0%)	–	–
M, n (%)	M0	29 (67%)	9 (64%)	0.75 ^F^	1.24 (0.27–5.15) ^OR^
	M1	13 (30%)	5 (36%)		
	Mx	1 (2%)	0 (0%)	–	–
Endocrine activity, n (%)	0	40 (93%)	12 (86%)	0.59 ^F^	2.19 (0.17–21.53) ^OR^
	1	3 (7%)	2 (14%)		
Cardiovascular diseases, n (%)	0	18 (42%)	10 (71%)	0.070 ^F^	0.30 (0.06–1.23) ^OR^
	1	25 (58%)	4 (29%)		
Diabetes, n (%)	0	29 (67%)	10 (71%)	>0.99 ^F^	0.83 (0.16–3.57) ^OR^
	1	14 (33%)	4 (29%)		
Tumors (other), n (%)	0	38 (88%)	14 (100%)	0.32 ^F^	0.00 (0.00–3.37) ^OR^
	1	5 (12%)	0 (0%)		
*Family history*					
Breast, ovarian, prostate, or pancreatic cancer, n (%)	0	25 (58%)	7 (50%)	>0.99 ^F^	1.38 (0.35–5.55) ^OR^
	1	18 (42%)	7(50%)		
Colorectal cancer, n (%)	0	31 (72%)	10 (71%)	>0.99 ^F^	1.03 (0.20–4.53) ^OR^
	1	12 (28%)	4 (29%)		
Other tumors, n (%)	0	28 (65%)	9 (64%)	>0.99 ^F^	1.03 (0.23–4.24) ^OR^
	1	15 (33%)	5 (36%)		

^F^ Fisher’s exact test, ^W^ Wilcoxon rank-sum test, ^FFH^ Fisher–Freeman–Halton test. ^OR^ odds ratio, ^d^ Cohen’s d, ^φc^ Cramer’s V correlation.

**Table 2 curroncol-32-00631-t002:** Pathogenic, VUS, and benign variants along with clinical details: grading, staging, age at diagnosis, and years of observation.

Variant Type	Detected Variant	Diagnosis	Age at Diagnosis	Years of Observation
Pathogenic variants				
1	*CHEK2* c.(908+1_909-1)_(1095+1_1096-1)del	G1, pT1N0M0	50	16
2	*BRCA2* c.8165C>G p.(Thr2722Arg)	G1, TXN1M1	70	16
Variants of uncertain significance				
1	*PMS2* c.2003T>C p.(Ile668Thr) VUS	G1, T1N0M0	43	5
2	*PMS2* c.2003T>C p.(Ile668Thr) VUS	G2, TXN1M1	60	14
3	*APC* c.3436C>T p.(Arg1146Cys) VUS	G1, T1N1M0	55	11
4	*MSH6* c.3242T>G p.(Leu1081Trp) VUS	G1, pT1N0M0	50	6
5	*PALB2* c.3122A>C p.(Lys1041Thr) VUS	G2, TXN1M1	60	14
6	*EPCAM* c.577A>G p.(Ile193Val) VUS	G2, TXN1M1	51	14
7	*MSH2* c.2178G>C p.(Met726Ile) VUS	G2, T2N0M0	65	11
8	*CHEK2* c.470T>C p.(Ile157Thr) VUS	G2, T1N0M0	58	5
9	*CHEK2* c.470T>C p.(Ile157Thr) VUS	G2, T1N1M1	29	1
10	*CHEK2* c.470T>C p.(Ile157Thr) VUS	G2, TXN1M1	55	5
11	*CHEK2* c.470T>C p.(Ile157Thr) VUS	G2, pT3N0M0	65	7
Benign variants				
1	*STK11* c.1208A>G p.(Lys403Arg)	G1, pT1N0M0	35	9
2	*APC* c.4919G>A p.(Arg1640Gln)	G1, TXN1M0	36	9
3	*APC* c.4919G>A p.(Arg1640Gln)	G1, T1N1M0	55	11
4	*BRCA2* c.7448G>A p.(Ser2483Asn)	G2, TXN1M1	60	14
5	*BRCA2* c.10095_10096insT p.(Ser3366Ter)	G2, pT2N0M0	65	9

## Data Availability

The data underlying this study consist of sensitive patient information and are therefore subject to strict confidentiality, GDPR regulations, and ethical approval restrictions. In accordance with these legal and ethical obligations, the datasets generated and analyzed during the current study cannot be made publicly available or shared with third parties under any circumstances. This ensures full compliance with patient privacy protection and the conditions imposed by the institutional ethics committee.

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
