# Peer review of "Germline Mutations in DNA Repair Genes in Patients with Pancreatic Neuroendocrine Neoplasms: Diagnostic and Therapeutic Implications"

_curroncol, 2025, doi:10.3390/curroncol32110631_

Round 1

Reviewer 1 Report

Comments and Suggestions for Authors

Dear Authors,

Thank you for presenting a well-written and informative manuscript on an emerging disease such as Pancreatic Neuroendocrine Neoplasms, which represent the most common type of pancreatic cancer after PDAC. Although my overall assessment of the manuscript is positive, I have a few minor comments and questions that I believe could further strengthen your work.

  1. As the paper primarily focuses on Pancreatic Neuroendocrine Neoplasms, it might be beneficial to include a brief section summarizing key aspects of the disease, such as its development, cellular origin, and prevalence.
  2. I would appreciate more details regarding the inclusion criteria for patient selection. Specifically, why were only well-differentiated tumors of grades 1–3 included? By excluding patients with grade 4 or poorly differentiated tumors, potentially valuable data on an important subset of the population - relevant for therapeutic decision-making - may be overlooked.
  3. Could you elaborate on the rationale behind the selection of markers for targeted sequencing? It might also be useful to mention known biomarkers commonly applied in PDAC screening in the introduction. In this context, including BRCA1 (in addition to BRCA2) and other genes involved in DNA maintenance, such as XRCC4/6 and FANCM, would enhance the comprehensiveness of your analysis.

Minor comment:

  • Figure 1: The figure legend should be clarified, as several abbreviations are missing.

Author Response

Response to Reviewer

We sincerely appreciate the reviewer’s time and effort in evaluating our manuscript. Below, we provide detailed responses to each comment, with all corresponding revisions and corrections clearly marked in the resubmitted files. We are grateful for the constructive feedback, which has helped us improve the quality and clarity of the paper.

Comment 1:

As the paper primarily focuses on Pancreatic Neuroendocrine Neoplasms, it might be beneficial to include a brief section summarizing key aspects of the disease, such as its development, cellular origin, and prevalence.

Response 1:

We thank the Reviewer for this valuable comment. We fully agree that including a concise summary of the key aspects of Pancreatic Neuroendocrine Neoplasms (pNENs) would enhance the clarity and completeness of our manuscript. Accordingly, we have revised the Introduction section to include an overview describing their cellular origin, development, and prevalence.

Comment 2:

I would appreciate more details regarding the inclusion criteria for patient selection. Specifically, why were only well-differentiated tumors of grades 1–3 included? By excluding patients with grade 4 or poorly differentiated tumors, potentially valuable data on an important subset of the population—relevant for therapeutic decision-making—may be overlooked.

Response 2:

We appreciate the Reviewer’s insightful comment. Only well-differentiated pNENs (G1–G3) were included to ensure a biologically and histologically homogeneous cohort, consistent with the study’s aim and prior genomic frameworks such as Scarpa et al.Nature, 2017 (ICGC). As pNENs are rare tumors, focusing on this subgroup allows meaningful aggregation of comparable cases and facilitates integration of our data with existing international datasets. Poorly differentiated NECs harbor distinct molecular features (e.g., TP53, RB1 alterations) and would require separate analytic and clinical stratification. This limitation and the need for future studies including NECs have now been acknowledged in the revised Discussion.

Additionally, this has been specified in section 4.3. Limitations, where we added the following clarification:

“Given the rarity of pNENs, focusing on well-differentiated cases (G1–G3) allows meaningful aggregation and comparison of genomic findings across international datasets, including the ICGC reference cohort by Scarpa et al., thereby enhancing the interpretability and translational relevance of the results.”

Comment 3:

Could you elaborate on the rationale behind the selection of markers for targeted sequencing? It might also be useful to mention known biomarkers commonly applied in PDAC screening in the introduction. In this context, including BRCA1 (in addition to BRCA2) and other genes involved in DNA maintenance, such as XRCC4/6 and FANCM, would enhance the comprehensiveness of your analysis.

Response 3:

We thank the Reviewer for this pertinent and detailed comment. To address it clearly and comprehensively, our response has been divided into two parts.

  1. Rationale for the selection of markers used in targeted sequencing
    The selection of genes for targeted sequencing was guided by previously published genomic studies, particularlyScarpa et al.,Nature, 2017 (ICGC), to ensure comparability and enable integration of our findings with larger international datasets. Given the rarity of pNENs, adopting a gene profile consistent with prior research facilitates aggregation of biologically similar cases and strengthens the molecular and clinical interpretability of the results. Moreover, several of the selected genes are known to have potential therapeutic or predictive significance, particularly in the context of DNA damage response and targeted treatment strategies.

This clarification has been included in section 2.6. Molecular Analysis:

“The gene panel used in this study was designed based on previously published large-scale genomic analyses of pancreatic neuroendocrine neoplasms (e.g., Scarpa et al.Nature, 2017; ICGC), allowing meaningful comparison across rare tumor cohorts and supporting both molecular interpretation and potential therapeutic relevance.”

  1. Consideration of additional genes relevant to PDAC (e.g., BRCA1, XRCC4/6, FANCM)
    We appreciate the Reviewer’s thoughtful suggestion. However, the present study focused specifically on germline variants recurrently observed in well-differentiated pNENs, based on prior ICGC andScarpa et al.data. While BRCA1 and other DNA maintenance genes (XRCC4/6, FANCM) are relevant in PDAC, they were not included here because a pathogenic BRCA2 variant identified in our cohort represents the first and only molecular feature linking pNENs to PDAC biology. Expanding the panel to additional PDAC-related genes is planned for future research.

This clarification has been included in section 4. Discussion:

“Although several genes such as BRCA1, XRCC4/6, and FANCM are established biomarkers in pancreatic ductal adenocarcinoma (PDAC), they were not included in the present analysis, as the pathogenic BRCA2 variant identified in our cohort represents the first and only molecular feature linking pNENs to PDAC biology.”

Minor Comment:

Figure 1: The figure legend should be clarified, as several abbreviations are missing.

Response:

We thank the Reviewer for this helpful comment. The legend of Figure 1 has been revised and expanded to include all relevant abbreviations for improved clarity.

Additional Clarifications

The revised manuscript has been sent to MDPI Author Services for comprehensive editing to ensure full compliance with journal formatting standards and reviewer recommendations. In particular, figures (especially Figure 1) and tableswill be corrected to meet MDPI’s visual and structural requirements, and the overall layout of the manuscript will also be adjusted accordingly.

We are currently awaiting the corrected version of the figures, tables, and layout from Author Services, which should be delivered within 24 hours. Once these revised materials are received, the complete corrected manuscript will be resubmitted to MDPI.

Reviewer 2 Report

Comments and Suggestions for Authors

The article by Beata Jurecka-Lubieniecka and colleagues, “Germline Mutations in DNA Repair Genes in Patients with Pancreatic Neuroendocrine Neoplasms: Diagnostic and Therapeutic Implications,” is an interesting and well-done pilot study. The work looks at how common germline DNA repair gene mutations are in 57 Polish patients with pNENs and what that might mean for diagnosis and treatment. Using targeted NGS, they found pathogenic or uncertain variants in around 25% of the cohort – mainly in CHEK2 and BRCA2. This is quite a meaningful observation suggesting that even apparently sporadic pNENs may have underlying hereditary components. I think the message about importance of germline testing for such cases is clear and valuable.

Overall, the paper is clearly written and methodologically sound, though there are few areas where improvement could make it stronger and a bit more informative for the readers.

Major Comments

  1. Scope and Novelty 
    The study touches on an important but still under-explored area. The regional focus (Polish cohort) adds value but I think novelty can be better demonstrated by deeper comparison with global germline datasets to see allele frequency and variant pattern differences.
  2. Methodological Claity        
    Methods are well described but justification for selecting those specific 11 genes should be added. Also mention sequencing depth/coverage and QC thresholds so that analytic rigor is clear.
  3. Comparative / Functional Correlation     
    Currently only basic clinical variables are compared. It would add more weight if authors could correlate mutation carriers with response or outcomes (platinum, PARPi, immunotherapy etc.).
  4. Figures and Data Presentation      
    Figure 1 needs higher resolution and better labels. Supplementary file with variant table + ACMG codes would make it more transparent.
  5. Therapeutic Implications   
    Discussion mentions PARPi and immunotherapy in general, but data is not presented. A small “Clinical Translational Outlook” section summarizing future perspective will make it complete.

Minor Comments

  • Abstract and simple summary are fine but please ensure same numbers (like 14/57) appear throughout.
  • In introduction, reference to NCCN or ESMO germline testing guidelines would situate the work.
  • Keep protein notation consistent (e.g. p.Thr2722Arg not T2722R).
  • References should be in MDPI superscript style.
  • Mention the software (with version) in stats section, and significance threshold.
  • Some background sentences in Discussion repeat from Introduction – can be shortened.
  • Define all abbreviations on first use (VUS, MMR, DSB) and fix small typos like “en-hancers” → “enhancers”.

Author Response

Response to Reviewer 

We sincerely appreciate the reviewer’s time and thoughtful evaluation of our manuscript. The comments provided were highly constructive and have helped us substantially improve the clarity, structure, and scientific value of the work. Below, we address each point in detail, with all corresponding revisions incorporated into the resubmitted version.

Comment 1: Scope and Novelty

The study touches on an important but still under-explored area. The regional focus (Polish cohort) adds value but I think novelty can be better demonstrated by deeper comparison with global germline datasets to see allele frequency and variant pattern differences.

Response 1:

Thank you very much for this insightful comment regarding the need for a deeper comparative analysis with global germline datasets. A comparative analysis was performed between the Polish cohort and global germline datasets. In response to your suggestion, we have incorporated an appropriate section in the revised manuscript.
The following paragraph has been added to the Introduction section:

“The frequency of the CHEK2 del5395 variant in the Polish group (2%) is similar to values reported for the Finnish European population (2.576%), while that variant is far less common—below 0.1%—in African American and East Asian populations [https://gnomad.broadinstitute.org/]. Comparable population-specific patterns are observed for other tested genes, which might highlight the unique genetic landscape of the Polish cohort and its relevance for cancer risk assessment.”
In addition, a new table presenting allele frequencies of individual variants across different populations has been created and added to the supplementary materials.

Comment 2: Methodological Clarity

Methods are well described but justification for selecting those specific 11 genes should be added. Also mention sequencing depth/coverage and QC thresholds so that analytic rigor is clear.

Response 2:

Thank you very much for your comment. We fully agree that providing justification for the selected gene panel and including details on sequencing coverage and quality control enhances methodological transparency.
The following clarification has been added to section 2.6. Molecular Analysis:

“The selected 11 genes (BRCA1, BRCA2, PALB2, CHEK2, MLH1, MSH2, MSH6, PMS2, EPCAM, APC, MUTYH) were carefully chosen based on their well-documented role in DNA repair mechanisms, particularly via homologous recombination and mismatch repair (MMR) pathways, which are crucial for maintaining genomic stability. This panel is routinely used in genetic diagnostics, which further supports its clinical relevance.”
Furthermore, additional details on sequencing coverage, software versions, and quality thresholds have been added to the Bioinformatics Analysis section to ensure analytic rigor.

Comment 3: Comparative / Functional Correlation

Currently only basic clinical variables are compared. It would add more weight if authors could correlate mutation carriers with response or outcomes (platinum, PARPi, immunotherapy etc.).

Response 3:

We appreciate the reviewer’s insightful comment. However, current therapeutic guidelines for pancreatic neuroendocrine neoplasms (pNENs) (ENETS 2023, ESMO 2024, NCCN 2025) do not include platinum-based chemotherapy, PARP inhibitors, or immune checkpoint inhibitors as part of standard management. These approaches remain investigational and are currently being tested in a limited number of early-phase clinical trials (e.g., NCT04593713, NCT04921240, NCT05058651).
Therefore, correlation between mutational status and response to these agents could not be assessed in our cohort, as none of the patients received such treatments outside of experimental settings.
To address this point, a clarifying note has been added in the newly created section “Clinical Translational Outlook” in the Discussion, emphasizing the current research efforts and therapeutic implications of these molecular findings.

Comment 4: Figures and Data Presentation

Figure 1 needs higher resolution and better labels. Supplementary file with variant table + ACMG codes would make it more transparent.

Response 4:

We thank the Reviewer for this valuable comment. The legend of Figure 1 has been revised to include full definitions of all abbreviations. The figures were submitted at a resolution of 800 DPI, which exceeds the standard 300 DPI requirement, ensuring high image quality. Therefore, the issue may have arisen during the editorial display or formatting process.
In addition, a Supplementary File (to Figure 1) has been prepared, containing the variant table with corresponding ACMG classification codes, to ensure full transparency and facilitate interpretation.

Comment 5: Therapeutic Implications

Discussion mentions PARPi and immunotherapy in general, but data is not presented. A small “Clinical Translational Outlook” section summarizing future perspective will make it complete.

Response 5:

We thank the Reviewer for this insightful suggestion. In response, a new subsection entitled “Clinical Translational Outlook” has been added to the Discussion, summarizing the therapeutic implications of the findings and outlining potential directions for future clinical research.

Minor Comments

Abstract and simple summary are fine but please ensure same numbers (like 14/57) appear throughout.
Response: We thank the Reviewer for this remark. The entire manuscript has been carefully reviewed, and all numerical data have been cross-checked. We confirm that there are no discrepancies in the reported numbers throughout the text, including the Abstract and Simple Summary.

In introduction, reference to NCCN or ESMO germline testing guidelines would situate the work.
Response: The authors appreciate the reviewer’s valuable suggestion. Appropriate clarifications and references to the current NCCN and ESMO germline testing guidelines have been added to the Introduction to better contextualize the study within current international recommendations.

Keep protein notation consistent (e.g. p.Thr2722Arg not T2722R).
Response: We appreciate the reviewer’s attention to detail. The entire manuscript has been carefully reviewed, and all protein variants are consistently presented according to the HGVS nomenclature (e.g., p.Thr2722Arg).

References should be in MDPI superscript style.
Response: The authors acknowledge the reviewer’s remark. The manuscript will be submitted to the journal’s Author Services for formatting adjustments, including the conversion of references to the required MDPI superscript citation style and verification of all other technical requirements.

Mention the software (with version) in stats section, and significance threshold.
Response: Thank you very much for this valuable suggestion. We appreciate the attention to detail regarding software versioning and reporting statistical rigor. In response, we have complemented the Bioinformatics Analysis section of the manuscript with information on the exact versions of all programs used in the computational pipeline.
“Sequence reads were aligned to the human reference genome using the Burrows–Wheeler Aligner (v 0.7.17) and Spliced Transcripts Alignment to a Reference aligner (v 2.7.11). Variant calling was performed using the Genome Analysis Toolkit (v 4.0.1.2) HaplotypeCaller and ExomeDepth. Variants were annotated and classified in accordance with American College of Medical Genetics and Genomics (ACMG) guidelines using Local Run Manager and Variant Interpreter (Illumina Inc., v 2.16). The threshold of coverage was set at 20x for small variants (SNP, small indels) and 30x for CNV. Quality check was performed according to Illumina recommendation using Sequencing Analysis Viewer software (v 2.4.7).”

Some background sentences in Discussion repeat from Introduction – can be shortened.
Response: We appreciate the reviewer’s valuable observation. The Discussion section has been carefully revised to remove or shorten repetitive background sentences and to focus on the interpretation of the results. The overlapping statements with the Introduction have been streamlined to avoid redundancy while preserving clarity and context.

Define all abbreviations on first use (VUS, MMR, DSB) and fix small typos like “en-hancers” → “enhancers”.
Response: We appreciate the reviewer’s careful observation. All abbreviations have been defined at first use (VUS, MMR, DSB), and minor typographical issues (e.g., hyphenation in “enhancers”) have been corrected throughout the manuscript.

Additional Clarifications

The revised manuscript has been sent to MDPI Author Services for comprehensive editing to ensure full compliance with journal formatting standards and reviewer recommendations. In particular, figures (especially Figure 1) and tables will be corrected to meet MDPI’s visual and structural requirements, and the overall layout of the manuscript will also be adjusted accordingly.
We are currently awaiting the corrected version of the figures, tables, and layout from Author Services, which should be delivered within 24 hours. Once these revised materials are received, the complete corrected manuscript will be resubmitted to MDPI.